# Does CABG with Saphenous Vein Grafting and Standard Cardiac Rehabilitation Affect Lower Limb Function? A Clinical Study

**DOI:** 10.3390/ijerph16111903

**Published:** 2019-05-29

**Authors:** Aleksandra Skomudek, Grzegorz Waz, Krystyna Rozek-Piechura

**Affiliations:** 1Faculty of Clinical Physiotherapy, Department of Exercise Science and Physiotherapy, Opole University of Technology, ul. Proszkowska 76, 45-758 Opole, Poland; 2Medinet Heart Disease Clinic of Lower Silesia, ul. Kamienskiego 73A, 51-124 Wrocław, Poland; gwaz12@gmail.com; 3Department of Physiotherapy, University of Physical Education, al. I. J. Paderewskiego 35, 51-612 Wroclaw, Poland; krystyna.rozek-piechura@awf.wroc.pl

**Keywords:** lower limb, CABG, saphenous vein, plethysmography, thermography, Biodex

## Abstract

Background: The aim of the study was to assess lower limb function in response to two cardiac rehabilitation (CR) protocols after coronary artery bypass surgery with saphenous vein grafting. Methods: Clinically-stable male patients aged 50–70 years were recruited 4 weeks post-surgery in which to group. Group I (*n* = 47) receive standard CR in a hospital setting for 3 weeks and Group II (*n* = 14) receive CR with a resistance training component in an outpatient setting for 8 weeks. Measures included body mass and composition, lower limb temperature distribution, lower limb hemodynamics, and dorsal and plantar flexor muscle strength. Results: Average temperature of the operated limb decreased only in Group II after cardiac rehabilitation. Venous blood flow improved in both groups as evidenced by increased blood refilling time. Isokinetic strength was greater in Group I. Conclusions: The results suggest a 3-week intensive CR protocol to be most effective in restoring lower limb function in CABG patients after saphenectomy.

## 1. Introduction

While coronary artery bypass grafting (CABG) remains the most widely used treatment modality for coronary artery disease (CAD), the procedure is associated with several cardiac and non-cardiac complications. Post-operative cardiopulmonary health is severely compromised with impaired pulmonary and respiratory function mediated by such factors as chest wall pain, decreased respiratory muscle strength, reduced exercise capacity and exercise tolerance, and anxiety with the resumption of more vigorous physical activity [1,2,3].

Supervised exercise-based cardiac rehabilitation (CR) has been strongly recommended in recent years as an important adjunct therapy following CABG with well-documented benefits [4,5,6,7]. CR is associated with improved prognosis after revascularization and reduced first year mortality and morbidity with improvements in such critical patient health outcomes as cardiorespiratory function, exercise tolerance, mental health, and quality of life [7,8,9]. Previous studies have investigated various exercise prescriptions regarding mode of activity, intensity, duration, frequency, training modality, and progression with emphasis that the inpatient and outpatient phases of CR involve sufficient exercise volume as well as that they be individually personalized due to clinical and physiologic diversity [7,10,11,12,13]. While most usual-care CR in bypass patients is based on aerobic training, an increasing body of literature has recommended the concomitant application of lower limb resistance training to enhance the beneficial effects of CR including increased skeletal muscle strength (particularly lower limb muscle volume), an important peripheral factor influencing cardiac function [12,14,15]. 

While the benefits of CR on increasing mobility and functional strength are known [7,16], no data is available on the specific effects of CR on lower limb function following CABG. This is particularly relevant as the great saphenous vein is commonly harvested for coronary revascularization and therefore may be a source of additional post-operative complications. Furthermore, there is a paucity of evidence on CR-induced changes in venous function after CABG with little clarity on other vascular adaptations of the gastrocnemius muscle after saphenous vein harvesting. This is particularly important as the calf muscle pump is an important hemodynamic component of the venous system as it forces the return of venous blood to the heart from the lower limbs, thereby preventing blood pooling and venous reflux [17,18]. 

Isokinetic limb testing is commonly used in clinical and research settings for the measurement of lower limb strength and power in patients with CAD, chronic lower limb ischemia, or after myocardial infarction [19,20,21]. Testing is performed on an isokinetic dynamometer to measure maximal muscle contraction force at predetermined angular velocities, most commonly at 60°/s, 180°/s, and 240°/s [19,20,22]. In addition to isokinetic testing, recent studies have adopted thermography [23,24,25,26,27,28,29] and photoplethysmography [27,30,31,32] as low-cost and non-invasive techniques to measures vascular function and changes in lower limb blood volume, respectively. An integrated assessment of both temperature distribution and hemodynamics is credited with enhancing diagnosis of inflammation and other disorders and in the monitoring of healing processes [30,32,33,34]. The aim of the study was to assess lower limb function in CABG patients with saphenous vein grafts after two different phase II cardiac rehabilitation protocols measuring lower limb temperature distribution, lower limb hemodynamics, and dorsal and plantar flexor muscle strength in isokinetic conditions.

## 2. Materials and Methods

The study was supported by the National Science Center with a Preludium 5 Panel NZ 7 grant (UMO-2013/09/N/NZ7/03650). The study protocol was registered with the Australian New Zealand Clinical Trials Registry (ACTRN12615000983505) and approved by the Senate Ethics Committee for Scientific Research of the University of Physical Education in Wroclaw, Poland (Ethic Project Identification code: 26/2013). 

Study population: A total of 120 male CAD patients were selected from two different hospitals 21–24 days after CABG via sternotomy using saphenous vein grafts. Patients in the first hospital (Group I) were to receive standard CR in a hospital setting for 3 weeks whereas patients in the second hospital (Group II) were to attend 8 weeks of CR with a resistance training component in an outpatient setting. Inclusion criteria were: age 50–70 years, clinically stable with no post-operative complications, and eligibility for enrollment in phase II CR. The exclusion criteria were: renal or liver dysfunction; inflammation; acute bronchitis; pneumonia; tuberculosis; asthma; skin disease at measurement site, any mental, orthopedic, neurological impairments or dysfunction that may impede participation in the study; and involvement in rehabilitative treatment 1 year prior to study outset. All patients provided their informed consent to participate in the study. 

Following the application of the inclusion and exclusion criteria, Group I consisted of 47 patients and Group II consisted of 14 patients. Participant flow and retention is illustrated in Figure 1 with final group characteristics provided in Table 1. Patients in Group I and Group II were then qualified to an appropriate exercise program (B or C) based on the results of an electrocardiographic ramp-based stress test [4] and consultation with the patient’s physician (Table 2). All tests were performed under medical supervision. 

CR in Group I was involved a regime of gymnastics, general aerobic exercise, and ergometer cycling held twice a week. Breathing exercises were performed twice a day. In addition to the above, endurance training was administered on a cycle ergometer twice a week in which pedaling frequency was changed every 20–30 s. Exercise intensity for all training modalities was individually prescribed based on the results of the graded exercise test. Outpatient CR in Group II involved interval training on a cycle ergometer three times per week and a general and resistance training workout twice a week. Interval training involved ergometer cycling for 4 min followed by 2 min of low-intensity cycling at 0–5 W. Initial training intensity was individually determined based on the results of the initial stress test. Workload was gradually increased in the first half of each training session until peak power was reached and then steadily decreased. Starting intensity was increased by no more than 10 W every fourth session during the first half of intervention [35]. Each session was preceded by a warm-up and, upon completion, a cool-down by cycling for 3 min with no load.

### 2.1. Study of Temperature Distribution in Lower Limbs

Lower limb temperature distribution was measured from the medial ankle to the knee. Minimum, maximum, and average temperature was determined with the use of a Variocam infrared camera (Infratec, Germany) from a distance of 150 cm. All measurements were taken in a temperature-controlled room (ca. 24 °C and 45% humidity) and in accordance with the recommendations of the European Association of Thermology. Temperatures were recorded across three trials, each involving 25 images taken at a frequency of one image per second. Average temperature (Tavg) and the difference in maximum and minimum temperature (ΔTmax–Tmin) was used for analysis. In order to ensure proper working conditions of the camera, the device was turned on ~30 min before imaging.

### 2.2. Dynamics of Venous Blood Flow in Lower Extremities

Lower limb hemodynamics were assessed by measuring venous refilling time (RT) and venous pump power (VP) with a Rheo Dopplex II photoplethysmography (Huntleigh Diagnostisc, UK). The device probe was attached to the skin 10 cm above the medial malleolus and 1 cm posterior. The patient sat in a chair (feet flat on the floor and knees bent at approximately 110°) and executed 10 dorsiflexions first with the one and then the second limb. The patient executed the 10 dorsiflexion rhythmically for 45 s and then kept the leg stationary for another 45 s during which three measures of RT and VP were taken and then averaged.

### 2.3. Study of Muscle Torque of the Lower Extremities in Isokinetic Conditions

Lower limb muscle function was quantified by measuring isokinetic calf muscle function with a Multi-Joint 4 dynanometer (Biodex, USA). The patient performed a series of cyclical ankle plantarflexions/dorsiflexions in the sagittal plane at 60°/s (5 repetitions) and then at 120°/s (10 repetitions). Measures included plantar flexor and dorsal flexor peak torque, peak torque to body mass, deficit peak torque, total work done, and average power.

### 2.4. Statistical Analyses

Data were processed with the Statistica PL software package (StatSoft, Poland). The distribution of the data set was screened for normality using the Shapiro–Wilk test and the mean values were compared using analysis of variance (ANOVA) for repeated measures and then analyzed post hoc with Fisher’s Least Significant Difference (LSD) test. All data is presented at mean ± standard deviation. Statistical significance was set at 0.05 and significant results are reported in bold. 

## 3. Results

### 3.1. Lower Limb Temperature Distribution

In both groups, pre-CR comparisons of Tavg revealed the operated limb to have a significantly higher mean temperature than the non-operated limb (Table 3 and Table 4). After CR was administered, Tavg decreased significantly only in Group II. However, the difference in Tavg between the operated and non-operated limb was still significant in Group I and Group II. No significant differences were observed before CR for ΔTmax–Tmin in either group nor were any significant pre- and post-intervention differences found.

### 3.2. Lower Limb Hemodynamics 

Before CR, RT was significantly lower in the operated limb than the non-operated limb in both groups. After CR, RT of the operated limb significantly increased by a similar magnitude in both. VP was slightly lower in the operated limb in both groups although this difference did not achieve statistical significance. However, post-CR VP of the operated limb increased significantly in Group II (Table 3 and Table 4).

### 3.3. Lower Limb Muscle Function 

Pre- and post-CR isokinetic measures are presented in Table 5 and Table 6. Prior to CR, almost all plantar- and dorsiflexion measures were significantly lower for the operated limb in Group I whereas only plantarflexion at 60°/s was lower in Group II (no differences were observed for any of the dorsiflexion measures). After CR, significant and similar increases in the majority of the plantar- and dorsiflexion measures were observed in both groups. Lower limb function was largely restored with little differences between the operated and non-operated limb in Group I whereas significant differences between the two limbs were still found in Group II.

In Group I, significant post-CR increases were observed in all measures for the operated limb. For the non-operated limb, significant increases were found only for peak torque, peak torque to body mass, total work done, and average power at 60°/s plantarflexion. In Group II, pre- and post-CR comparisons for the non-operated limb revealed a significant increase in all plantar- and dorsiflexion measures at 60°/s and for all dorsiflexion measures at 120°/s. For the operated limb only two changes achieved statistical significance: a decrease in plantarflexion total work done at 120°/s and an increase in dorsiflexion average power at 60^o^/s. When deficit peak torque was analyzed, significant post-CR decreases were observed in both dorsi- and plantarflexion movements at 60°/s in Group I whereas in Group II only deficit peak torque decreased during plantarflexion compared with a post-CR increase in dorsiflexion. A similar trend was observed in deficit peak torque at 60°/s although none of these differences were significant (Table 5 and Table 6). 

## 4. Discussion

CR is a critical component of post-operative therapy for CAD patients [35,36,37]. Longitudinal research has found lower limb strength to be compromised after cardiac surgery compared with pre-operative levels but steadily improves with values reaching close to baseline at hospital discharge [38]. The results of the present study confirm the benefits of CR, particularly within the domain of lower limb function after CABG with saphenous vein harvesting. We observed an evident reduction in function of the operated limb after CABG (pre-CR) when compared with the non-operated limb. However, a significant improvement was found in isokinetic strength at post-CR with fewer differences between the operated and non-operated limb. A review of the literature found only one related study which assessed changes in limb function following CABG although the radial artery was harvested [29]. In this study, no changes were observed in either handgrip strength or hand temperature although this may be explained by the study’s less-invasive harvesting technique of a smaller blood vessel and also the non-demanding nature of the testing protocol. 

Regarding the two cardiac rehabilitation protocols that were assessed, we found Group II (less-intensive 8-week program) to show an enhanced reduction in operated-limb average temperature than Group I (intensive 3-week program). This may be explained by the duration of the intervention particularly in regard to the amount of time provided for post-operative recovery and the total training volume of the two interventions, in which Group I exercised almost daily compared with Group II that exercised only three times per week.

Analysis of lower limb hemodynamics in both groups revealed RT was significantly lower in the operated limb and increased upon concluding CR. A different scenario was observed with VP, in which the only significant difference was a post-CR increase in Group II. While no similar works on post-cardiac surgery lower limb hemodynamics could be found, a study on the effects of Nordic walking on venous blood flow via photoplethysmography in older females reported enhanced lower limb RT when compared with an untrained control group, indicating the positive effects of physical activity on blood flow dynamics [39].

The literature is more abundant on assessing lower limb function via isokinetic testing in patients with cardiovascular disease [19,21]. We observed a significant increase in the majority of the isokinetic strength measures of the operated limb in both groups with a slightly enhanced effect in Group I. The results confirm the findings of other studies, such as the positive benefits of an 8-week aerobic and resistance program on lower limb isokinetic strength [22] or 1-month CR on lower limb strength and static and dynamic balance [40].

It is important to note that the majority of CABG patients after saphenous vein grafting complain of lower limb complications (many due to infection) rather than those at the sternal wound site [41]. Infection at the saphenous vein harvest site is modulated by such risk factors for infection as obesity, female sex, diabetes, peripheral vascular disease, smoking, preoperative anemia, chronic renal failure, patients with an intra-aortic balloon pump, and operative technique. While skin-related symptoms tend to be minimal, they can cause significant morbidity including cellulitis, peripheral sensory deficit, and vein graft dermatitis [41]. After saphenous vein harvesting, the operated limb has an increased tendency to develop swelling and local complications due to damaged capillaries, impaired venous drainage, and trauma to lymphatic and soft tissue. While the present group of CABG patients were free of wound complications at the harvest site, we nonetheless observed a reduction in lower limb function compared with the non-operated leg. Continuing this important line of research, future studies should involve patients at greater risk of complications, as well as those with post-operative morbidity related to saphenous vein harvesting as well as assess additional quality of life markers. Moreover, the authors are strongly considering the continuation of studies in patients who underwent endoscopic harvesting of the long saphenous vein.

## 5. Conclusions

Decreased function of the operated limb was observed at pre-CR when compared with the non-operated limb, indicating the adverse effects of saphenous vein harvesting.

Post-CR comparisons between the operated and non-operated revealed that there were still significant differences for average temperature and venous refilling time after 3-week CR (Group I) and 8-week CR (Group II), although significant increases in these parameters were noted. Isokinetic testing revealed improved lower limb function after 3-week CR compared with 8-week CR, with fewer differences between the operated and non-operated limb.

While both interventions significantly improved lower limb function in this group of patients, the 3-week protocol with greater exercise frequency was more effective in reducing the pre-CR differences between the operated and non-operated limb.

## Figures and Tables

**Figure 1 ijerph-16-01903-f001:**
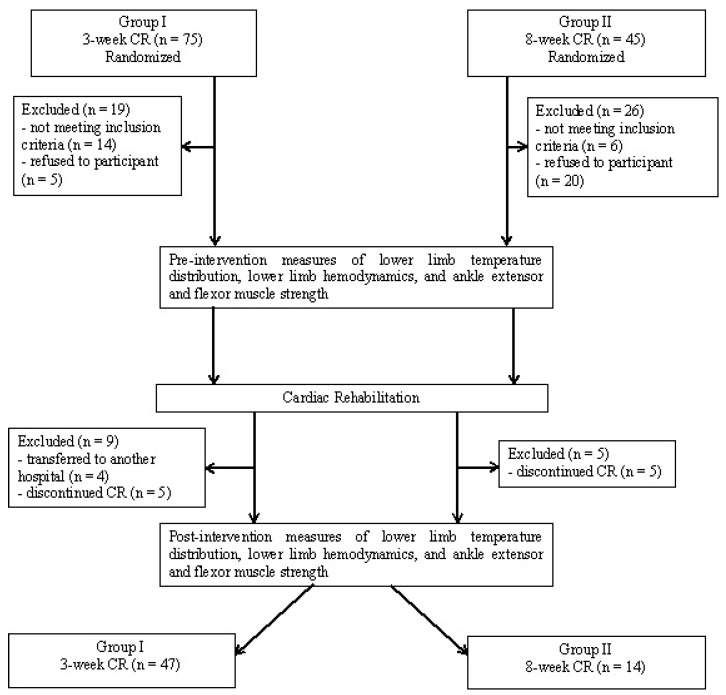
Participant flow diagram.

**Table 1 ijerph-16-01903-t001:** Group characteristics.

Variable	Group I *n* = 47		Group II *n* = 14		LSD Post-Hoc Significance of Differences
	$\bar{x}$	*SD*	$\bar{x}$	*SD*	*p*
Age (years)	59.51	7.06	63.14	8.46	0.1118
Body height (cm)	172.09	5.63	175.43	5.23	0.0524
Body mass (kg)	83.32	12.46	88.72	16.56	0.1856
BMI (kg/m^2^)	28.06	3.34	29.04	4.86	0.3874

**Table 2 ijerph-16-01903-t002:** Participants allocated to CR protocol B or C.

Variable	Group		
	Group I *n* = 47	Group II *n* = 14	*p*
CR protocol (Price et al., 2016)			
B	*n* = 23 (48.94%)	*n* = 7 (50%)	0.9454
C	*n* = 24 (51.06%)	*n* = 7 (50%)	

**Table 3 ijerph-16-01903-t003:** Pre- and post-CR lower limb temperature distribution and hemodynamics.

	Studies							
	Pre-CR				Post-CR			
	Group I		Group II		Group I		Group II	
	OP	NOP	OP	NOP	OP	NOP	OP	NOP
T_avg_ (°C)	31.85 ± 0.95	32.28 ± 1.06	30.52 ± 1.18	30.81 ± 0.67	31.82 ± 1.15	32.09 ± 1.20	29.82 ± 1.19	31.09 ± 1.03
∆ (T_max_–T_min_) (°C)	3.13 ± 1.14	3.39 ± 1.02	3.53 ± 0.87	3.30 ± 1.05	2.85 ± 1.03	3.41 ± 1.20	3.19 ± 1.08	3.36 ± 0.92
RT (s)	21.87 ± 9.16	28.03 ± 11.60	20.59 ± 9.89	29.64 ± 8.34	25.50 ± 10.94	29.58 ± 11.25	24.70 ± 10.09	29.78 ± 7.27
VP	30.04 ± 10.31	31.35 ± 11.82	28.14 ± 9.42	32.38 ± 11.84	31.31 ± 11.34	31.01 ± 11.30	35.02 ± 14.57	31.14 ± 10.81

T_avg_—average temperature, Δ(T_max_–T_min_)—temperature differential, RT—refilling time, VP—venous pump power.

**Table 4 ijerph-16-01903-t004:** Comparisons of pre- and post-CR lower limb temperature distribution and hemodynamics.

	*p* Value								
	Operated–Non-Operated Limb				Pre–Post CR				Post CR
	Group I		Group II		Group I		Group II		I-II
	Pre-CR	Post-CR	Pre-CR	Post-CR	OP	NOP	OP	NOP	OP
T_avg_ (°C)	**0.0015**	**0.0401**	**0.0419**	**0.0000**	0.1523	0.8303	**0.0041**	0.2397	**0.0000**
∆ (T_max_–T_min_) (°C)	**0.0004**	0.0519	0.3660	0.5210	0.8889	0.0570	0.1951	0.8064	0.5573
RT (s)	**0.0000**	**0.0046**	**0.0007**	**0.0500**	**0.0114**	0.2683	**0.0411**	0.9546	0.8239
VP	0.9792	0.5554	0.3788	0.3185	0.4378	0.8311	**0.0243**	0.6773	**0.0463**

Notes: Data in bold indicates statistical significance at *p* < 0.05. T_avg_—average temperature, Δ(T_max_–T_min_)—temperature differential, RT—refilling time, VP—venous pump power.

**Table 5 ijerph-16-01903-t005:** Lower limb muscle function (ankle dorsiflexion/plantarflexion) at pre-cardiac rehabilitation and post-cardiac rehabilitation.

Variable	Angular velocity (^o^/s)	Flexion	Pre CR				Post CR			
			Group I		Group II		Group I		Group II	
			OP	NOP	OP	NOP	OP	NOP	OP	NOP
Peak Torque	60	Plantar	33.19 ± 15.28	39.46 ± 20.33	25.80 ± 13.55	31.97 ± 12.99	43.65 ± 19.59	44.75 ± 18.68	32.79 ± 15.28	34.00 ± 19.70
		Dorsi	19.31 ± 7.42	21.23 ± 6.16	24.96 ± 9.74	24.93 ± 10.05	21.86 ± 6.16	22.28 ± 6.58	25.96 ± 10.95	29.73 ± 9.40
	120	Plantar	26.08 ± 12.20	29.51 ± 15.03	23.47 ± 12.40	26.79 ± 10.19	29.77 ± 13.12	29.60 ± 12.70	26.44 ± 12.92	27.06 ± 12.41
		Dorsi	13.84 ± 5.44	15.51 ± 5.26	17.70 ± 8.23	19.18 ± 7.05	15.36 ± 4.73	15.07 ± 4.17	19.20 ± 7.90	23.39 ± 7.14
Peak TQ/BM	60	Plantar	40.91 ± 18.55	48.47 ± 24.17	29.51 ± 11.68	38.90 ± 15.06	53.24 ± 23.92	54.88 ± 23.68	38.84 ± 20.45	39.19 ± 17.38
		Dorsi	23.66 ± 8.64	25.67 ± 6.95	29.13 ± 14.13	29.84 ± 12.51	26.46 ± 8.10	26.91 ± 7.76	30.98 ± 13.95	34.49 ± 13.13
	120	Plantar	32.20 ± 13.95	36.07 ± 17.33	27.60 ± 10.97	31.52 ± 12.43	36.34 ± 16.14	36.38 ± 16.07	30.62 ± 12.21	32.30 ± 10.58
		Dorsi	16.75 ± 6.55	18.82 ± 6.05	20.80 ± 10.36	22.80 ± 8.63	18.67 ± 5.75	18.54 ± 5.70	22.60 ± 10.25	26.94 ± 8.36
Total Work	60	Plantar	76.31 ± 47.16	94.12 ± 62.37	66.71 ± 34.45	81.91 ± 56.54	103.27 ± 57.98	108.08 ± 61.97	73.85 ± 44.43	85.39 ± 41.98
		Dorsi	46.62 ± 24.03	54.40 ± 23.79	73.59 ± 39.96	78.49 ± 40.21	57.76 ± 24.22	59.53 ± 25.05	78.78 ± 41.89	93.69 ± 46.49
	120	Plantar	85.84 ± 12.52	124.52 ± 63.10	67.21 ± 17.96	135.34 ± 57.42	118.92 ± 70.13	112.44 ± 71.01	113.25 ± 65.71	130.39 ± 67.73
		Dorsi	55.70 ± 27.71	67.15 ± 31.15	100.20 ± 48.90	105.88 ± 54.92	69.06 ± 27.83	71.74 ± 33.41	103.94 ± 61.10	124.19 ± 47.89
Avg Power	60	Plantar	17.08 ± 10.43	18.35 ± 11.54	11.82 ± 6.64	15.49 ± 7.10	22.94 ± 11.05	24.10 ± 12.87	16.33 ± 8.79	17.19 ± 9.26
		Dorsi	10.88 ± 4.99	11.11 ± 4.09	14.07 ± 6.16	14.44 ± 6.33	11.64 ± 3.69	12.74 ± 4.46	16.25 ± 7.31	18.39 ± 6.43
	120	Plantar	19.22 ± 10.64	19.56 ± 12.40	17.02 ± 9.72	19.46 ± 8.04	21.27 ± 11.05	22.96 ± 12.53	18.92 ± 9.02	19.49 ± 10.40
		Dorsi	9.98 ± 4.14	10.04 ± 5.29	15.08 ± 7.02	15.59 ± 8.70	10.27 ± 3.46	11.27 ± 3.98	17.39 ± 8.28	20.55 ± 6.99

**Table 6 ijerph-16-01903-t006:** Probabilities for post-hoc LSD comparisons of pre- and post-intervention operated and non-operated limb muscle function during plantarflexion and dorsiflexion at 60°/s and 120°/s.

Variable	Angular Velocity (°/s)	Flexion	*p* Value								
			Operated–Non-Operated Limb				Pre–Post CR				Post
			Group I		Group II		Group I		Group II		I–II
			Pre-CR	Post-CR	Pre-CR	Post-CR	OP	NOP	OP	NOP	OP
Peak Torque	60	Plantar	**0.0003**	0.4996	**0.0427**	0.6850	**0.0000**	**0.0019**	**0.0078**	0.7855	0.1028
		Dorsi	**0.0174**	0.5979	0.9840	**0.0103**	**0.0004**	0.4223	**0.0013**	0.4807	0.2020
	120	Plantar	**0.0007**	0.8600	**0.0441**	0.7284	**0.0003**	0.9315	**0.0467**	0.8812	0.5822
		Dorsi	**0.0182**	0.6687	0.2452	**0.0015**	**0.0306**	0.5223	**0.0014**	0.2385	0.0775
Peak TQ/BM	60	Plantar	**0.0005**	0.4311	**0.0162**	0.9253	**0.0000**	**0.0030**	**0.0169**	0.9387	0.0769
		Dorsi	**0.0220**	0.5963	0.6500	**0.0286**	**0.0003**	0.3621	**0.0011**	0.4712	0.2616
	120	Plantar	**0.0023**	0.9722	**0.0391**	0.6876	**0.0012**	0.7965	0.1801	0.7279	0.3903
		Dorsi	**0.0128**	0.8667	0.1817	**0.0048**	**0.0204**	0.7254	**0.0070**	0.2287	0.1410
Total Work	60	Plantar	**0.0002**	0.2795	**0.0242**	0.3220	**0.0000**	**0.0024**	**0.0446**	0.1584	0.1522
		Dorsi	**0.0079**	0.5332	0.9562	**0.0003**	**0.0002**	0.0744	**0.0048**	0.3203	0.1673
	120	Plantar	0.3648	**0.0228**	**0.0440**	0.9425	**0.0001**	0.3418	0.9425	**0.0390**	0.2065
		Dorsi	**0.0034**	0.4779	0.7787	**0.0009**	**0.0007**	0.2262	**0.0100**	0.5893	0.0315
Avg Power	60	Plantar	0.2710	0.3146	**0.0420**	0.6785	**0.0000**	**0.0002**	**0.0121**	0.6885	0.0577
		Dorsi	0.6477	**0.0271**	0.6842	**0.0193**	**0.0003**	0.2804	**0.0000**	**0.0462**	0.0585
	120	Plantar	0.6973	0.0580	0.1324	0.7257	**0.0003**	**0.0224**	0.1291	0.7357	0.3388
		Dorsi	0.9246	0.1147	0.6606	**0.0078**	0.0550	0.6535	**0.0000**	0.1226	0.0029

Notes: Data in bold indicates statistical significance at *p* < 0.05.

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
