# Peer review of "Does CABG with Saphenous Vein Grafting and Standard Cardiac Rehabilitation Affect Lower Limb Function? A Clinical Study"

_ijerph, 2019, doi:10.3390/ijerph16111903_

Round 1
Reviewer 1 Report
Line 22: In the prior line you mentioned 75. However, the #s in each group don’t appear to add up to 75
Line 75: grammar issue: …by based… (needs revision)
Line 95: to clarify, was there randomization OR was allocation only a result of hospital guidelines?
Line 106: the “held twice week” requires revision for grammar.
Line 108: what is “ca.”?
Line 125: could benefit from grammar revision. The part “entailed the right and left medial from medial ankle ..” is confusing
Line 142: maybe try: …execute the 10 dorsiflexion movements rhythmically…
Line 184: tables 5 and 6
Author Response
We are very thankful to the reviewers for their time and effort. Their comments have greatly improved our article and we hope our answers and changes in the manuscript will prove to be satisfactory, especially regarding how our
Kind Regards
The Authors
Reviewer 1:
Comments and Suggestions for Authors
Line 22: In the prior line you mentioned 75. However, the #s in each group don’t appear to add up to 75
Thank you, this was corrected in the manuscript
Line 75: grammar issue: …by based… (needs revision) –
This was corrected.
Line 95: to clarify, was there randomization OR was allocation only a result of hospital guidelines? –
Due to a translation error this was incorrectly formulated. The groups were recruited from two separate hospitals and this was deliberate, as one hospital administered only the 3-week cardiac rehabilitation and the other only the 8-week program. The entire manuscript was updated to correctly state our study design. To better show how the participants were chosen and then retained through the study, we created a participant flow diagram and included it in the manuscript (Figure 1).
Line 106: the “held twice week” requires revision for grammar.
Corrected in the manuscript.
Line 108: what is “ca.”?
This is an abbreviation for circa about but was deleted.
Line 125: could benefit from grammar revision. The part “entailed the right and left medial from medial ankle ..” is confusing
This was corrected in the manuscript.
Line 142: maybe try: …execute the 10 dorsiflexion movements rhythmically…
Much better, this was corrected in the manuscript
Line 184: tables 5 and 6
Corrected in the manuscript

Reviewer 2 Report
I personally appreciate for this kind of clinical study, however there are few basic points where I need clarification.
Please explain in detail the type of study design? Its create confusion like sampling method, process of recruitment in both group etc.
Is it any kind of blinded study either from the side of accessor or after training?
Please explain the reason why you recruit unequal sample in both arm?
Is this is a feasibility study or comparative study?
Author Response
We are very thankful to the reviewers for their time and effort. Their comments have greatly improved our article and we hope our answers and changes in the manuscript will prove to be satisfactory, especially regarding how our
Kind Regards
The Authors
Reviewer 2:
Comments and Suggestions for Authors
I personally appreciate for this kind of clinical study, however there are few basic points where I need clarification.
Please explain in detail the type of study design? Its create confusion like sampling method, process of recruitment in both group etc.
Agreed, we can now see the confusion as due to a translation error this made it appear as if we had one population that was than separated into two groups. We compared patients from two hospital CR programs. The first hospital only administered 3-week CR (Group I) whereas the second hospital administered only 8-week CR (Group II) and the patients we recruited were from these two, separate groups. This error has now been corrected with the necessary changes throughout the manuscript.
Is it any kind of blinded study either from the side of accessor or after training?
The authors and individuals administering the pre- and post-tests were not blinded but the patients and physiotherapists in charge of therapy were blinded to the purpose of the study.
Please explain the reason why you recruit unequal sample in both arm?
The unequal samples were due to the reduced number of available participants in Group II, and this can be attributed to how the CR programs were administered. Patients in Group I were operated on and then continued to CR within the hospital setting. Patients in Group II however performed CR in an outpatient setting and had to systematically attend all sessions on their own. We created a participant flow diagram to better show the entire recruitment/inclusion/retention process in the manuscript (Figure 1).
Is this is a feasibility study or comparative study?
A comparative study, where we compared two different CR protocols to determine if a longer duration outpatient model is more effective than a shorter inpatient model.

Reviewer 3 Report
Overall the article confirms the effectiveness of the traditional approach.
Would the authors consider to continue the study in patients who underwent endoscopic harvesting of the long saphenous vein? This may well shed some more lights on the controversy between open and endoscopic technique and ascertain if endoscopic harvesting is really beneficial in reducing postoperative complications and improve function.
Author Response
We are very thankful to the reviewers for their time and effort. Their comments have greatly improved our article and we hope our answers and changes in the manuscript will prove to be satisfactory, especially regarding how our
Kind Regards
The Authors
Reviewer 3:
Comments and Suggestions for Authors
Overall the article confirms the effectiveness of the traditional approach.
Would the authors consider to continue the study in patients who underwent endoscopic harvesting of the long saphenous vein? This may well shed some more lights on the controversy between open and endoscopic technique and ascertain if endoscopic harvesting is really beneficial in reducing postoperative complications and improve function.
We wholeheartedly agree and included this comment at the end of our Discussion that future research should include patients who underwent endoscopic harvesting.